# Cell Immortality: *In Vitro* Effective Techniques to Achieve and Investigate Its Applications and Challenges

**DOI:** 10.3390/life14030417

**Published:** 2024-03-21

**Authors:** Mahla Chalak, Mahdi Hesaraki, Seyedeh Nasim Mirbahari, Meghdad Yeganeh, Shaghayegh Abdi, Sarah Rajabi, Farhid Hemmatzadeh

**Affiliations:** 1Department of Developmental Biology, School of Basic Sciences and Advanced Technologies in Biology, University of Science and Culture, Tehran P.O. Box 871-13145, Iran; 2Department of Stem Cells and Developmental Biology, Cell Science Research Center, Royan Institute for Stem Cell Biology and Technology, ACECR, Tehran 1665659911, Iran; 3Department of Photo Healing and Regeneration, Medical Laser Research Center, Yara Institute, ACECR, Tehran 1417613181, Iran; 4Department of Genetics, Reproductive Biomedicine Research Center, Royan Institute for Reproductive Biomedicine, ACECR, Tehran 1665659911, Iran; 5School of Biotechnology, College of Science, University of Tehran, Tehran 1417935840, Iran; 6Department of Cell Engineering, Cell Science Research Center, Royan Institute for Stem Cell Biology and Technology, ACECR, Tehran P.O. Box 16635-148, Iran; 7Australian Centre for Antimicrobial Resistance Ecology, School of Animal & Veterinary Sciences, University of Adelaide, Roseworthy Campus, Roseworthy, SA 5371, Australia

**Keywords:** immortalization, cell division, telomeres, *hTERT*, cell line

## Abstract

Cells are very important to researchers due to their use in various biological studies in *in vitro* and *in vivo* settings. This importance stems from the short lifespan of most cells under laboratory conditions, which can pose significant challenges, such as the difficulties associated with extraction from the source tissue, ethical concerns about separating cells from human or animal models, limited cell passage ability, and variation in results due to differences in the source of the obtained cells, among other issues. In general, cells in laboratory conditions can divide into a limited number, known as the Hayflick limit, due to telomere erosion at the end of each cellular cycle. Given this problem, researchers require cell lines that do not enter the senescence phase after a limited number of divisions. This can allow for more stable studies over time, prevent the laborious work associated with cell separation and repeated cultivation, and save time and money in research projects. The aim of this review is to summarize the function and effect of immortalization techniques, various methods, their advantages and disadvantages, and ultimately the application of immortalization and cell line production in various research fields.

## 1. Introduction

In 1965, Leonard Hayflick introduced the idea that cells possess a mechanism for keeping track of the number of times they divide, a concept later termed the Hayflick limit, which aligns with the number of replication cycles [1]. The molecular basis of this phenomenon is based on the gradual shortening of telomeric DNA (Figure 1). The shortening of telomeres occurs with each cell division, ultimately causing cells to reach their Hayflick limit, whereby growth stops after approximately 60 times population doubling [2,3,4,5]. When telomeres become too short and cannot function naturally, cellular senescence in the M1 phase (a cellular growth arrest, also called the M1 stage) occurs [5,6,7]. Thus, very short telomeres are identified by cells as double-stranded breaks and create DNA damage responses that include cellular apoptosis and replicative senescence mechanisms [4].

Therefore, due to the short lifespan of cells in laboratory conditions, a continuous supply of these cells from their specialized sources seems necessary for scientific research. However, the repeated procurement of cells requires much skill and experience if the donor is available, as mature cells from their origin tissues (such as vascular tissue) are difficult to obtain [8]. 

Furthermore, the preparation of primary human and animal tissues for the extraction and isolation of various cell types is fraught with significant challenges, including the potentially invasive nature of the process, an increased risk of infection, pain, ethical issues, and the difficulty of the process itself. In some cases, some patients may not consent to the use of excess samples separated from their organs, which can delay access to cellular resources in research. In addition to all these problems, obtaining cells from various sources and their availability for conducting a study can lead to errors and changes in the interpretation and results of research. For example, as human umbilical vein endothelial cells (HUVECs) are obtained from the umbilical cord, they are not only influenced by maternal hormones but also affected by embryonic gonads. In other words, even the fetal sex from which HUVEC cells are isolated can play a significant role in the reproducibility of the results. In fact, various studies have shown sexual differences in the umbilical cord between men and women in terms of gene expression, protein expression, cell survival, tubule formation capacity, autophagy, cellular ATP and metabolite levels, oxidative stress, and angiogenesis [9,10,11,12]. This challenge can also apply to other cells. Therefore, researchers need immortalized cells that do not enter the senescence phase after several cell divisions, considering the limitations of obtaining more adaptable cells over time.

Immortalized cell lines are modified cells that can be grown indefinitely [13]. Ideally, immortalized cells are genetically and phenotypically similar or identical to their source tissue and can reproduce and be cultured in the long term. Immortalized cell lines can be used in most research instead of primary cells because they offer several advantages, including cost-effectiveness, ease of use, unlimited availability of materials, and the bypassing of ethical concerns associated with animal and human tissue use [14,15,16].

In the late 1950s, George Otto Gey, head of tissue culture research at Johns Hopkins, initiated the process of immortalization and established a non-aging cell line called HeLa, derived from a rare adenocarcinoma found in the cervix of a young woman named Henrietta Lacks. This marked the first instance of such a phenomenon [17,18]. Prior to this, cells derived from other human cells could only survive for a few days in culture, but the behavior of Lacks’s tumor cells was different. This was the first successful attempt to immortalize and maintain human cells *in vitro* because the ability of Hela cells made them useful in various biological studies [19,20,21,22,23]. 

In fact, HeLa cells have an active version of telomerase during cell division [24], which copies telomeres repeatedly. This prevents the gradual shortening of telomeres, which plays a role in cell aging and, ultimately, cell death. Thus, cells evade the Hayflick limit, resulting in unlimited cell division and immortality. Major discoveries have been made using this cell line, including the development of the polio vaccine in 1953 [25], the link between human papillomavirus (HPV) and cervical cancer, and the role of telomerase in chromosome maintenance [26]. 

Generally, various cell lines have revolutionized scientific investigations and have found applications in vaccine development, drug testing for metabolism and toxicity, generating antibodies, exploring gene functions, creating artificial tissues like synthetic skin, and producing biologic substances such as therapeutic proteins [16,19,26]. The aim of this review is to address the process of immortalization of cells and various methods of creating different cell lines and their use in research.

## 2. The Application of Immortalized Cell Lines

Immortalized cell lines are a powerful tool for biological, biochemical, and biological growth, differentiation, and aging studies. They are also used in immunology, hematology, cancer biology, and toxicology research. Additionally, for therapeutic purposes, studying immortalized cells will be useful in achieving better results for regenerative medicine [27]. Cell line immortalization can also help cell biologists achieve their goals of treating diseases and improving health factors. For example, they have used immortalized cell lines as model systems for studying neuronal development and performance recovery in neurological disease models, such as Huntington’s disease [28,29].

In general, the use of these cell lines can be useful for clinical development related to the treatment of Huntington’s disease patients [29]. Another example is human vocal fold epithelial cells, which are a valuable tool for studying epithelial–fibroblast cell interactions that dictate the disease and health of this specialized tissue [30]. The RPE-1 human retinal pigment epithelial cell line has also been widely used to study physiological events in human cellular culture systems [31]. Pig endothelial cells can be also used as a laboratory model to study the properties of the blood–brain barrier and some hemorrhagic diseases [32]. Encapsulated cell technology is also a useful approach for the continuous and local delivery of genetically modified therapeutic proteins. However, the clinical development of encapsulated cell technology to deliver therapeutic proteins from macro-capsules is still limited due to the lack of a compatible allogeneic cell line for therapeutic purposes in humans [33].

In this regard, it is possible to create appropriate immortal cell lines, such as the human myoblast cell line created in the study by Lathuiliere A. and colleagues. Valuable results have also been achieved using enclosed cell technology. As an example, non-immortalized human lung epithelial cells created by *hTERT* overexpression without the use of viral oncogenes have been used to investigate various aspects of lung cancer, such as epithelial-to-mesenchymal transition and the cancer stem cell theory. The use of non-immortalized lung epithelial cells has improved researchers’ understanding of lung cancer pathogenesis, and these models can be valuable research tools [34]. Additionally, creating cell lines that preserve genetic information and drug responses can enable more drug screening and mechanistic studies [35]. 

The immortalized NTera2 (NT2) cell line has undergone a thorough examination for its potential in brain grafting. Notably, these cells have been safely employed in clinical trials focused on treating brain injuries [36,37]. Lately, a method involving the lentiviral gene transfection of *c-MYC* has been employed to create immortalized megakaryocyte cell lines, regulated by the TET-on system, using blood cell precursor cells derived from iPSCs (induced pluripotent stem cells). These cells have the capability to produce platelets for use in clinical trials [38]. Cell lines that are not immortalized can serve as valuable tools for investigating the effectiveness and potential harm of drugs. For instance, human cell lines have been extensively employed in pharmacogenomic research concerning cancer, aiding in the forecast of clinical responses, contributing to the development of pharmacogenomic theories for subsequent experiments, and uncovering fresh insights into the various factors influencing drug responses. Within the category of model system cell lines, immortalized types like EBV-transformed lymphoblastoid cell lines (LCLs) are frequently utilized to assess how genetic diversity impacts the effectiveness and safety of drugs [39].

Additionally, if we aim to apply this technique in a different research area, it would pertain to the creation of immortalized cell lines for food production, a significant hurdle in the field of cell agriculture. This field is dedicated to generating meat and other animal-derived products via tissue engineering and synthetic biology. Specifically, these cultured meat cell lines must meet criteria for food safety, be capable of large-scale propagation and differentiation in an efficient manner, and exhibit taste, texture, and nutritional qualities that appeal to consumers [40]. 

In this context, utilizing primary cells for cultured meat production necessitates the maintenance of donor animal herds for sample preparation, with regular and approved sample collection for food production. In contrast to primary cell cultures, immortalized cell lines do not undergo senescence and can undergo limitless divisions. As a result, they are more straightforward to investigate and enable the production of cultured meats that are both safer and more consistently achieved (Figure 2), eliminating the requirement for animal biopsies [27]. 

## 3. Telomeres

The important role of telomeres in ensuring chromosomal stability was first proposed in the 1930s by Barbara McClintock [41], who worked on corn, and Hermann Muller, who worked on fruit flies [42]. Both researchers suggested that the ends of chromosomes have special structures that are necessary for chromosomal stability. Muller coined the term telomere from the Greek words telos, meaning “end”, and meros, meaning “part”. Telomeres are necessary for protecting the ends of chromosomes, creating chromosomal stability, and ensuring the separation of genetic material into daughter cells after cell division [43,44]. 

Telomeres are nuclear protein complexes located at the ends of eukaryotic cell chromosomes [45]. This arrangement inhibits the identification of linear chromosome ends as double-stranded DNA breaks [46]. The DNA sequence of telomeres is characterized by repeated tandem sequences (TTAGGG) in all vertebrates (Figure 3) [47]. 

Telomeric DNA typically ends with a G-rich overhang, which consists of unpaired nucleotides at the 3’ end of a DNA molecule, ranging between 50 and 300 nucleotides [48]. The lengths of these repeats vary between chromosomes and species (Figure 3) [48,49,50].

In humans and mice, the length of telomere repeats at the ends of individual chromosomes varies significantly among cells. Human chromosome ends usually have detectable telomere repeats between 0.5 and 15 kilobases (kb), depending on factors such as tissue type, donor age, and cell replication history. Specific changes in telomere length occur at the ends of individual human chromosomes, with the average length varying between chromosome ends. For instance, chromosome 17p typically has shorter telomeres than other chromosome ends [51,52]. 

In human nuclear cells such as immune system cells, the average telomere length decreases significantly with age. The shortening of telomeres is a presumed tumor suppressor pathway mediated, in part, by the activation of cellular aging signals. The first scientist to link telomeres with planned cell division was Alexei Olovnikov [53]. He proposed that human somatic cells may not be able to correct chromosomal shortening that occurs during cell replication. He suggested that repeated nucleotide sequences in telomeres could act as a buffer to protect downstream genes from chromosome shortening in each cell division. Additionally, he had significant insights and suggested that the length of repeated sequences could determine the number of DNA replication cycles.

Watson also recognized that the nature of unidirectional DNA replication for complete end-to-end telomere replication poses a problem [54]. The first direct observations of telomeres’ direct association with aging in 1986 occurred when Cooke and Smith discovered that the average length of telomere repeats covering sex chromosomes in sperm cells is much greater than in adult cells [55]. They considered the possibility that adult cells may be deficient in the telomerase enzyme, which had recently been discovered in the single-cell organism Tetrahymena [56]. 

Several reports confirmed the decrease in average telomere length with cell division in fibroblasts [6,57,58] and with age in somatic blood and colon cells [59], but not in germ cells. These observations confirmed that somatic cells apparently cannot maintain telomere length, but with the use of telomerase reactivation techniques, it can be preserved to some extent. 

## 4. Immortality and a Review of Its Techniques

Numerous biotechnological approaches have been employed to manipulate the cellular genome in order to acquire immortalized cell lines. Nonetheless, the foremost and extensively applied techniques for achieving this involve the introduction of viral oncoproteins and telomerase reverse transcriptase (TERT) [60]. The immortality conferred by viral oncoproteins is intricately linked to the deactivation of cell cycle-regulating proteins (such as p16, p14, p21, p53, and Rb). Through this pathway, viral oncoproteins can induce the deactivation of tumor suppression mechanisms and may even trigger the expression of telomerase [61]. 

Furthermore, a correlation exists between the deactivation of pRb, cellular aneuploidy, and chromosomal instability. As a result, pRb and p53 emerge as pivotal proteins governing cellular senescence and replication. During senescence, an active, hyperphosphorylated form of pRb is present. This form binds with members of the E2F protein family, leading to the deactivation of certain genes involved in the G1/S phase transition. It is worth noting that Rb’s growth-inhibiting function appears to operate independently of p53. In human cells, p53 can initiate senescence autonomously without relying on Rb [62]. However, in general, the techniques used for cell immortalization by inactivating the two main tumor suppressor pathways, pRb and p53, can lead to genomic instability and the formation of polyploidy and altered chromosome numbers. Losing the function of the p53 gene can cause genomic instability and thus disrupt the ability of the genome to replicate accurately. This has been observed in about half of human malignant cancers [63] and underscores the vital role of p53 in tumor suppression. Therefore, the use of these techniques cannot be considered a safe and risk-free option for cell immortalization.

### 4.1. Viral Genes Can Control the Cell Cycle

As viruses rely on the ability to replicate within living organisms to survive, they are able to manipulate or accelerate the cellular cycle for their own benefit [64]. One of the methods to achieve this is through targeting the expression of Rb and p53 proteins. Viral oncogenes are also able to disable pRb and p53, thus overcoming the M1 barriers, which inhibit the growth and replication of natural cells, and significantly increasing the lifespan of cells [65].

#### 4.1.1. Simian Virus 40

The Simian Virus 40 (SV40) encodes two proteins, Large T antigen (LT) and Small T antigen (ST), which help induce virus-associated tumors. SV40 T antigen is currently used for cell transfection in different types of cells and can generate immortal cell lines by binding and disabling p53 and Rb proteins [66]. Both of these proteins have evolved to specifically target crucial cellular regulators and modify their functions. The Large T antigen, for instance, targets various known proteins encompassing 708 amino acids. These include three members of the retinoblastoma protein family (pRb, p107, and p130), as well as members of the Cap-binding protein (CBP) adaptor protein family, such as p300 and p400, along with the tumor suppressor protein p53. On the other hand, the Small T antigen affects the activity of the pp2A phosphatase and activates the cyclin A promoter. Notably, the LT protein plays a central role in conferring SV40’s extended lifespan, primarily due to its capability to interact with growth suppressors like pRb and p53 (Figure 4) [67] and suppress the p53 pathway [68]. 

Among the cell types that achieve immortalization through this mechanism, human proximal epithelial cells serve as an example, retaining their differentiation properties [69]. Conversely, Garcìa-Mesa and collaborators successfully immortalized microglial cells for the purpose of studying the latency and regulation of the human immunodeficiency virus (HIV) in the central nervous system (CNS). Importantly, they managed to preserve the majority of the primary glial cell’s phenotypic and functional characteristics [70]. Conversely, pre-adipocytes showed aberrant differentiation after immortalization with the SV40 T antigen. In fact, the ability of the SV40 T antigen to block the transcription factor p300/cAMP-response element-binding protein (CBP), which is essential for adipocyte differentiation, inhibits pre-adipocyte differentiation [71].

#### 4.1.2. Human Papillomavirus (HPV)

HPV is a small, double-stranded DNA virus that infects mucosal and cutaneous epithelial tissues [72]. High-risk strains, including HPV-8/16/18/31, cause malignant lesions, while low-risk strains, including HPV-6/11, cause benign warts and lesions [73]. The E6 and E7 proteins, encoded by high-risk strains such as HPV-16/18, are classified as oncoproteins [74]. When used in immortalization, E6 activates telomerase and accelerates the degradation of p53 by proteasome S26, while E7 can inactivate Rb by preventing the binding between pRb and the transcription factor E2F [75,76]. For example, Trakarnsanga and colleagues achieved the immortalization of early adult erythroblasts by employing HPV-16 oncoproteins. This led to the creation of stable cells capable of producing red blood cells [77]. 

Furthermore, by employing viral oncoproteins E6 and E7, along with the overexpression of the T SV40 antigen, epithelial cells located on the surface of the ovary have been successfully immortalized [78]. Overall, the transfer of certain cell types with E6/E7 HPV oncoproteins creates a cell line that retains many characteristics of primary tissue cells.

#### 4.1.3. Human T-Lymphotropic Virus (HTLV)

HTLV is a human RNA retrovirus that causes leukemia and adult T-cell lymphoma. There are two types, HTLV-1 and HTLV-2, which have the ability to infect lymphocytes under laboratory conditions, although clinically, HTLV-2 is less pathogenic than HTLV-1 [79]. Types 1 and 2 encode for the Tax1 and Tax2 proteins, respectively, which are necessary for the infection of human T-cells by the associated viruses. Due to the greater pathogenic power of HTLV-1, the number of growth-induced cells by Tax2 was much greater than the cells induced by Tax1, and the activity of Tax2 was far higher than Tax1 [80]. 

#### 4.1.4. Adenoviruses

Adenoviruses are common DNA viruses found in animals and humans and are often observed in adults and children [81]. The *12S E1A* gene product from adenovirus, belonging to the oncoprotein class, has the capability to establish primary cells as cell lines. It is encoded by two exons. Extensive mutational studies have revealed that four specific regions of the *12S E1A* gene, derived from both exons, play a critical role in prolonging the lifespan of primary epithelial cells. While the expression of two of these regions is essential for activating quiescent cells and initiating the cell cycle, it alone cannot confer immortality or extend their lifespan in culture. These two regions are encoded by exon 1. The third region within exon 1, whose function remains unidentified, is also indispensable for this process. Moreover, these three regions are crucial for cooperating with 12S and an activated *ras* gene in triggering tumor formation. The fourth region is necessary for sustaining the proliferation of cells, prolonging their lifespan in culture, and prompting autocrine growth factor production. Cells immortalized by both wild-type 12S and its mutant variants maintain their epithelial characteristics, and they continue to express intermediate filament proteins like keratin and vimentin [82]. For example, by transferring retroviral vectors containing coding sequences of *12S* or *13S E1A*, they caused the proliferation and immortalization of epithelial cells in primary cultures of the kidneys, liver, heart, pancreas, and thyroid of mice [83].

#### 4.1.5. Epstein–Barr Virus (EBV)

EBV is a double-stranded DNA virus that infects B lymphocytes. Immortalizing B cells is an effective method for inducing the long-term growth of some human B cells in laboratory conditions [84]. In fact, this virus can immortalize cells and convert them into lymphoblastoid cell lines that carry EBV [85]. These cells effectively induce specific T cell responses against EBV in laboratory conditions due to the presentation of viral antigens [86]. It has been shown that EBNA-2 is genetically essential for B cell immortalization by EBV. Experiments have shown that EBNA-2 affects the accumulation of viral and cellular RNAs, and LMP may also be necessary for immortalization as it can affect the growth properties of human lymphoid and epithelial cells. EBNA-1 may also be necessary for the immortalization of a B cell for EBV, as it appears to be necessary for the maintenance of viral DNA replication in the replicating cell population [87].

According to these results, we suggest that using viral genes for immortalization may affect the genome of target cells that can produce active oncogenes. So, this process is not recommended for the immortalization of primary cells. 

### 4.2. The Overexpression of Specific Genes for Immortalization

#### 4.2.1. The *HOX* Gene Family and *Lhx2*

The overexpression of the *HOX11* and *TLX1* genes can lead the hematopoietic precursor cells, mouse fetal liver, or bone marrow towards immortalization. A wide range of hematopoietic cell types have been immortalized using these methods, including erythroid, megakaryocytic, monocytic, myelocytic, and multipotent cells [88]. Studies have also shown that myelomonocytic, megakaryocytic, and mast cell progenitor immortalization in mice can be achieved using the *Hox-2.4* gene [89]. It has also been reported that the expression of the *HOXa9* gene can immortalize a promyelocyte. In fact, *Hoxa9* expression in primary murine marrows can immortalize a delayed myelomonocytic progenitor and prevent final differentiation into granulocytes or monocytes in the presence of granulocyte–macrophage colony-stimulating factor or interleukin-3 [90]. The expression of the *Lhx2* gene in mature bone marrow-derived hematopoietic stem/progenitor cells can provide the possibility of producing hematopoietic stem/progenitor cells dependent on sustained growth factor, which can produce erythroid, myeloid, and lymphoid cells after grafting into mice. Therefore, *Lhx2* is capable of immortalizing multipotent hematopoietic stem/progenitor cells, which can create functional outcomes after grafting into treated hosts [91]. 

#### 4.2.2. *c-Myc* Gene Expression

Another set of genes employed in the immortalization of cells belongs to the *myc* family. This family encompasses a group of oncogenes: *c-Myc*, *N-Myc*, *L-Myc*, and *B-Myc*. The expression of *c-Myc* is predominantly observed in actively proliferating cells, whereas *N-Myc* and *L-Myc* play roles in differentiation processes. Among these oncogenes, the c-Myc gene has been the subject of the most extensive studies [61]. It is worth noting that there is an association between p53 and c-Myc, as the Myc signaling pathway governs both apoptosis and cell immortalization, with the latter being contingent on the status of p53 [92]. 

While the overexpression of c-Myc leads to DNA damage and promotes genomic instability, it also circumvents the pro-apoptotic functions of p53 [93]. The excessive activity of c-Myc contributes to apoptosis by involving NF-κB mediation. Thus, the malfunctioning signaling of NF-κB is a necessary condition for Myc-induced carcinogenesis [94]. De Filippis and colleagues devised a technique to immortalize neural stem cells (NSCs) by introducing a retroviral vector carrying a mutated variant of *c-Myc* (c-Myc T58A) [95]. 

On the other hand, Li and colleagues also achieved a perpetual population of NSCs through L-Myc transduction [96]. Myc also stabilizes telomere length in human prostate epithelial cells (HPrECs) through the regulation of *hTERT* gene overexpression. Overall, HPrECs that are immortalized by the *c-Myc* gene maintain many normal cell characteristics, such as the induction of growth cessation in response to the Ras oncogene, intact p53 response, and absence of gross karyotypic abnormalities. However, they lack an Rb/p16INK4a surveillance checkpoint, which is a weakness of this method [97].

#### 4.2.3. *CDK4* Gene Expression

*CDK4* expression, along with increased *hTERT* gene expression, has been used for the immortalization of human bronchial epithelial cells [98]. Additionally, *CDK4* expression, along with cyclin D1 and increased telomerase activity, has been used for the immortalization of human myogenic cells derived from healthy and diseased muscles [99].

#### 4.2.4. *TERT* Gene Expression

As mentioned, the risk of oncogenic integration into chromosomes still raises various safety concerns when using an oncogenic transferable agent in host cells [61]. Using hTERT as a means of achieving immortalization through less phenotypic/karyotypic alterations has been proposed [60]. hTERT is a key determinant of human telomerase enzyme activity [100,101]. It comprises 16 exons and 15 introns and spans approximately 35 kb [102]. 

Telomerase is an enzyme responsible for maintaining telomere length [103] and was discovered in 1985 as an enzyme capable of extending telomeric repeat sequences. However, it was not until a decade later, in 1997, that the components of the telomerase protein complex were identified and thoroughly characterized [104]. This enzyme constitutes a sizable ribonucleoprotein complex tasked with the synthesis of telomeric DNA repeats in the forward direction. Broadly speaking, telomerase functions as a DNA polymerase and is composed of two distinct subunits: a catalytic-functional component called human telomerase reverse transcriptase (hTERT), encoded by the *TERT* gene, and an RNA component known as the human telomerase RNA component (hTERC or hTR), encoded by the *TERC* gene [5,102,105]. It is important to note that hTERC and hTERT are essential for reestablishing telomerase activity [106,107,108].

Overall, the *hTERT* gene produces a 1132 amino acid polypeptide that is then converted into a functional 130 kilodalton protein called TERT [109]. Four critical functional domains in TERT include the N-terminal regulatory domain, the RNA-binding domain, the reverse transcriptase domain, and the C-terminal dimerization domain [110]. Due to its complexity, TERT is regulated at various levels, including transcriptional, post-transcriptional, and post-translational mechanisms (Figure 4) [111,112,113]. Telomerase is primarily active in proliferating cells, hematopoietic stem cells, and rapidly renewing cells [114,115,116]. 

Conversely, in somatic cells, telomerase activity is minimal or non-existent, largely due to the tightly controlled regulation of hTERT [111]. The transfer of *hTERT* into human primary cells leads to an increase in telomere length and maintenance of chromosome ends. In many cases, the forced expression of *hTERT* alone enables cells to suppress aging in replication and overcome the growth crisis caused by telomere shortening [117,118,119]. Additionally, hTERT-immortalized cells exhibit physiological characteristics of normal cells inside the body and maintain phenotypic markers and stable karyotypes in high passages [120,121]. 

Human primary cells that have been immortalized by increasing hTERT expression in fibroblasts [122], choroidal melanocytes [31], endothelial cells [123], dermal keratinocytes, mammary epithelial cells, osteoblasts, and pancreatic cells [124].

Some important features observed in several types of immortalized cells by increasing hTERT expression include that they do not go towards malignancy [125,126], cell cycle control remains normal and p53 and pRb checkpoints remain active [125,127], contact inhibition is still normal [126], cells require growth factors for proliferation [118], and cell karyotyping remains normal and does not show extensive changes [119,127]. The shortening of telomeres can only be considered an anti-cancer mechanism if cell cycle control proteins (in checkpoint activities and telomere shortening) such as P53 and RB are working properly. hTERT-immortalized cells combine the physiological characteristics of primary cell lines and the continuous lifetime of cultured cell lines, preventing the aging process in primary cell lines and the unstable karyotype of cultured cell lines. Additionally, in many studies, hTERT-immortalized cells have been transformed into various differentiated cell types, exhibiting tissue-specific characteristics and unique proteins and forming structures similar to those inside the body [125]. 

In the field of overexpressing exogenes to immortalize cells, some genes have the ability to immortalize a series of specific primary cells (such as the *HOX* family or *c-Myc*, which have oncogenic activity), so using this method may affect the safety of the process. But using the *hTERT* gene for immortalization can be considered a safe method for immortalizing primary cells. A point that should not be forgotten is that *hTERT* is not in the category of oncogenes [116,128].

### 4.3. Cancer Cells and Immortalization

Cell immortalization is a crucial stage in tumorigenesis where cells can overcome aging and critical situations. Along this path, cells become immortal naturally and can be isolated. Tissue samples are acquired through a biopsy, followed by a process of separation, where various cell types are isolated and assessed for their proliferation potential and unique characteristics. This methodology formed the foundation for obtaining cell lines, commencing with the immortalized fibroblast cell line derived from mouse fibroblasts in the 1940s. Notably, the HeLa cell line, originating from cervical cancer, represents another significant milestone in this field [129,130]. 

However, in the process of the cell becoming cancerous, one of the contributing factors is cancer-associated fibroblasts or CAFs. CAFs are the dominant stroma surrounding the tumor. Studies have shown that CAFs can promote tumor growth, angiogenesis, resistance to chemotherapy and metastasis, and enhance cancer progression [131,132], thereby naturally directing cells towards immortality. A notable point in this regard is that the primary CAFs separated from human carcinomas have been shown to remain active even in laboratory conditions for a long period of time [133,134]. 

CAFs can be kept in a stable activation state to aid tumor malignancy. Epigenetic events can affect CAFs. DNA methylation is one of the important epigenetic changes (Figure 4). The reciprocal paracrine relationship between normal fibroblasts (NFs) and cancer cells leads to changes in DNA methylation in NFs, which plays a central role in NF planning for CAFs and CAF function. The presence of TGF-β1 as a primary cytokine is also essential for CAF activation. In fact, TGF-β1 derived from cancer cells can reduce miR-200 expression and, thus, help activate CAFs [135]. 

The expression of miR-200 is also concurrent with increased expression of the *DNMT3B* gene in CAFs. *DNMT3B* is not only a direct target of miR-221 but also affects miR-200b/c and regulates their expression in CAFs. On the other hand, miR-141 inhibition in these cells increases the expression of transcription factor 12 (TCF12) to facilitate the growth of breast cancer cells through CXCL12 secretion in CAFs, which leads to increased c-Myc and cyclin D1 expression in breast cancer cells, thus allowing these cells to undergo immortality through *c-Myc* activation [135]. 

In other words, these points indicate that the status of CAFs can be changed by modifying epigenetic factors and thus contribute to cell immortality as an example in the direction of proliferation. However, it is also essential to mention that cell immortalization cannot be considered a safe and reliable method in the laboratory to promote cell malignancy through epigenetic pathways.

### 4.4. Combining Methods

In some reports, the use of only one method may not be capable of producing non-dividing cells. Therefore, depending on different cell types, it is better to combine the cell cycle suppressor and hTERT expression to make more cells dividing. For example, human ovarian surface epithelial cells have been made non-dividing, providing good conditions for the study and progression of ovarian cancer. Inhibiting p53 expression with small interfering RNA through retroviral intermediaries can delay aging and increase cell division, but it is not enough to make natural ovarian surface epithelial cells non-dividing and human MSCs [136,137]. 

Merely introducing a subunit of the catalytic telomerase is inadequate in achieving a significant non-dividing state. However, the simultaneous suppression of p53 expression along with the overexpression of telomerase catalytic subunits proves to be sufficient in inducing cellular immortality in cultures of human ovarian surface epithelial cells. Additionally, WI-38 embryonic lung fibroblasts, widely used for studies on the aging process, have been made non-dividing with increased expression of the T antigen, *hTERT*, and *H-ras* genes, among other cases (Figure 4) [136]. In addition, WI-38 embryonic lung fibroblasts, which are widely used for studying the aging process, have been genetically modified to overexpress the T antigen, *hTERT*, and *H-ras* genes [138] and many other cases.

### 4.5. Chemical Components and Rays Role in Immortalization

Cells can be immortalized through exposure to radioactive factors and chemical carcinogens. Certain chemicals, some of which are carcinogenic, have the capacity to contribute to cell immortalization. These chemicals are categorized based on their carcinogenic potential. Each carcinogen can be further grouped based on its mode of action into genotoxic carcinogens [139] or non-genotoxic carcinogens [139]. Genotoxic carcinogens are substances or agents that directly initiate carcinogenesis by interacting directly with DNA, leading to DNA damage and chromosomal abnormalities that can be detected through genotoxicity testing. On the other hand, NGCs are agents capable of inducing cancer through a secondary mechanism, often as a result of their indirect impact on DNA. They have the ability to alter signal transduction pathways or gene expression. The GCs can be detected using genotoxicity testing, which detects changes to the cell at the molecular and cellular levels. These changes include mutations in genes, DNA strand breaks, the formation of DNA adducts, chromosomal aberrations, and aneuploidy [140]. As an illustration, potent mutagenic carcinogens like N-methyl-N-nitrosourea (MNU) and benzo(a)pyrene [141] have been demonstrated to be effective agents for immortalization in Syrian hamster dermal cells [142] cell transformation assay [143] through the direct inactivation of the tumor suppressors p53 and p16 (Figure 4) [144].

Physical carcinogens (such as ionizing radiation) are also powerful immortalization agents with different mechanisms and frequencies in rodent and human cells [140]. For example, X-rays, neutrons, and gamma rays produce immortal clones in SHD cells [144]. Conversely, the immortalization of human mammary cells through ionizing radiation is a relatively rare occurrence [145]. Similarly, methyl sulfate, a powerful clastogen, is an efficient immortalizing carcinogen in mammalian SDH cells and Chinese hamster cells [143] and has a similar mode of action to that of ionizing radiation.

Furthermore, the carcinogenic strength of basic aliphatic alkylating agents like alkylnitrosamides and alkylmethanesulfonates is directly associated with their capacity to modify the comparatively less reactive oxygen atoms in DNA, notably the O6 atom of guanine [143]. In addition there are data suggesting that acetaminophen activates telomerase [146,147,148,149], which could lead to the immortalization of cells. However, there are also data indicating that acetaminophen can inhibit CDK4 and CDK2, thus imposing a cell cycle checkpoint at G1 and effectively blocking cellular proliferation [140]. 

There are other data that show that aspirin, at therapeutically relevant concentrations, prevents the senescence of endothelial cells. This effect seems to be due to increased nitric oxide (NO) synthesis and reduced oxidative stress. Consistent with these findings, the formation of asymmetric dimethylarginine [150], an endogenous inhibitor of NO synthase, was decreased and the activity of dimethylarginine dimethylaminohydrolase (DDAH), an enzyme that degrades ADMA, was decreased [149].

### 4.6. Gene Manipulation and Immortalization

The CRISPR/Cas9 genome-editing system provides us an unprecedented opportunity to target and modify genomic sequences with high levels of efficacy and specificity [151,152,153,154,155]. The CRISPR/Cas9 system induces DNA double-strand breaks at specific sites of genomic DNA, which should allow a safer and more targeted gene delivery of the immortalizing genes [156]. Indeed, the most potent gene editing tool to date is encapsulated by the Clustered Regularly Interspaced Short Palindromic Repeats (CRISPR) and CRISPR-associated protein (CRISPR/Cas9) system [157]. 

This can be considered one of the most powerful and versatile forms of technology both for gene editing and transcriptional control but also for epigenetic modulation [158]. For example, in order to overcome the technical challenge of maintaining primary BMSCs in long-term culture, mouse bone marrow stromal stem cells (BMSCs), which are one of the most common mesenchymal stem cells, have been reversibly immortalized mediated CRISPR/Cas9, which can maintain the multipotent characteristics of mesenchymal stem cells (MSCs). In this study, by exploiting the CRISPR/Cas9-based homology-directed-repair (HDR) mechanism, they targeted SV40T to mouse Rosa26 locus and efficiently immortalized mouse BMSCs (i.e., imBMSCs) (Figure 4) [156].

Certainly, the study demonstrates that CRISPR-Cas9-mediated targeting of the p53 gene or CDKN2A locus proves to be an effective method for immortalizing primary marmoset skin fibroblasts. This is particularly significant as the common marmoset serves as a valuable non-human primate model for studying various human diseases. The research reveals that, similar to Cdkn2a-deficient mouse cells, CDKN2A-deficient marmoset cells express wild-type p53 proteins and respond normally to genotoxic stresses such as adriamycin and etoposide. These findings collectively underscore that Cas9-mediated gene targeting of the *p53* gene or *CDKN2A* locus is a potent tool for establishing immortalized marmoset cell lines with specific genetic modifications [159].

## 5. Challenges of Using Cell Lines

When opting for cell lines over primary cells, it is crucial to bear certain considerations in mind. Because cell lines undergo genetic manipulation, their phenotype, inherent functions, and responses to stimuli may undergo alterations. Another critical aspect to consider is the repeated cultivation of cell lines, which can bring about shifts in genotypic and phenotypic features over prolonged periods. This may result in genetic drift and heterogeneity within cultures and, consequently, may lead to variations in results. Consequently, cell lines may not entirely mirror all the attributes of primary cells in the long term and may yield differing outcomes. Two significant concerns related to cell lines are contamination by other cell lines and mycoplasma. Mycoplasma contamination, in particular, can persist undetected in cell cultures for extended periods, resulting in substantial alterations in gene expression and cellular behavior. This underscores the importance of vigilant monitoring and maintenance practices when working with cell lines [160,161]. Also, a 2011 study of 122 different head and neck cancer cell lines revealed that 37 (30%) were misidentified [162]. Analyses of a variety of tissue culture collections and cells sent to repositories for curation and storage from labs in the United States, Europe, and Asia suggest that at least 15% of cell lines are misidentified or contaminated [162,163].

In addition, a number of factors contribute to the problems of cell line misidentification and contamination. For example, inadvertently using a pipette more than once when working with different cell lines in culture can lead to cross-contamination. If the contaminating cell line divides more rapidly than the original cells, it can quickly dominate the population, changing the identity of the culture. This event often goes undetected because cells from different sources can be morphologically similar [164]. Walter Nelson-Rees brought to light a harsh reality in the early 1970s regarding the widespread cross-contamination of cell lines, both between different species and within the same species. He demonstrated that during that period, a significant portion of cell lines utilized globally and distributed by cell banks were tainted with HeLa cells. This revelation had profound implications for cell-based research and emphasized the need for rigorous quality control measures (Table 1) [165]. 

It is important to remember that cell lines may not exhibit the same behavior as primary cells. To bolster the validity of these results, it is essential to consistently conduct crucial control experiments using primary cells [172]. However, the potential challenge lies in the time and costs involved in conducting these tests. Nevertheless, based on the explanation provided, it is imperative to exercise caution when working with cell lines. It is strongly recommended to include experiments that validate key findings in primary cultures and to meticulously adhere to all relevant considerations.

## 6. Conclusions

Cell immortalization proves to be a valuable technique for acquiring cell lines with unrestricted replicative capacity. By following this approach, cells can surpass the Hayflick limit and evade the mechanisms linked to replicative senescence and eventual cell apoptosis. This is achieved through the upregulation of various viral oncogenes/oncoproteins, the TERT component of telomerase, and the expression of genes that regulate the cell cycle, ultimately leading to conditional immortality [61]. In addition, the use of immortal cell lines has been helpful in many researches, some of which were mentioned in this article, and these cases can present a sign of the importance of creating a cell line through the immortalization technique. It is worth noting that along with all the advantages that cell immortality provides for research, there are still problems (such as manipulation of the cell cycle when p53 and Rb change, chromosomal instability, etc.), and their solution requires more research in the future.

## Figures and Tables

**Figure 1 life-14-00417-f001:**
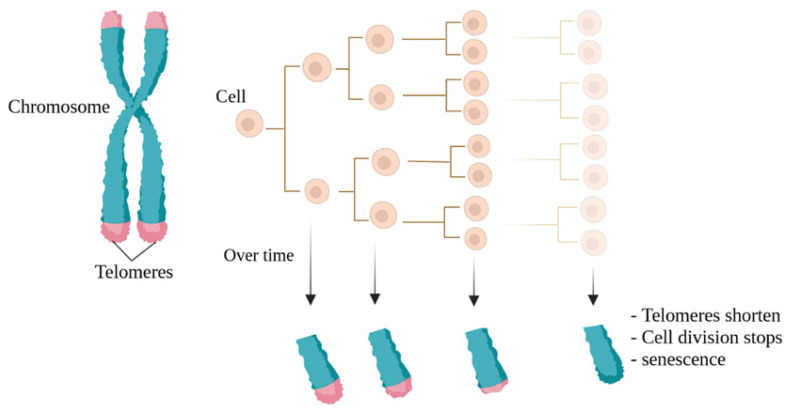
The Hayflick limit. Naturally, with each cell division, the telomeres at the ends of chromosomes become shorter, a phenomenon known as the Hayflick limit. The final stage is also known as cellular aging.

**Figure 2 life-14-00417-f002:**
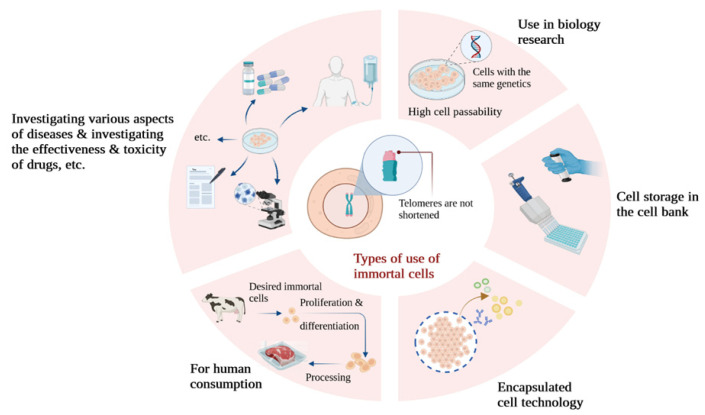
A look at some of the applications of immortalized cells in different types of research.

**Figure 3 life-14-00417-f003:**
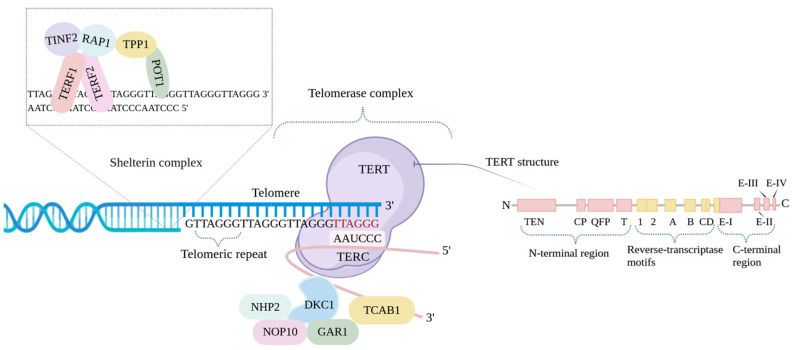
Telomerase components and TERT structure. Telomerase is composed of three main components—hTERT, hTERC, and DKC1. The hTERC is associated with DKC1 and small nucleolar RNPs, NOP10, NHP2, and GAR1. The sequence, by adding repetitive TTAGGG sequences to the end of the telomere, prevents its shortening and naturally prevents cell aging. The protein complex shelterin, or telosome, protects telomere ends. The shelterin complex is made up of telomeric repeat binding factors 1 and 2 (TERF1 and TERF2), repressor/activator protein 1 (RAP1), protection of telomeres 1 (POT1), TERF1 interacting nuclear factor 2 (TINF2), and TPP1.

**Figure 4 life-14-00417-f004:**
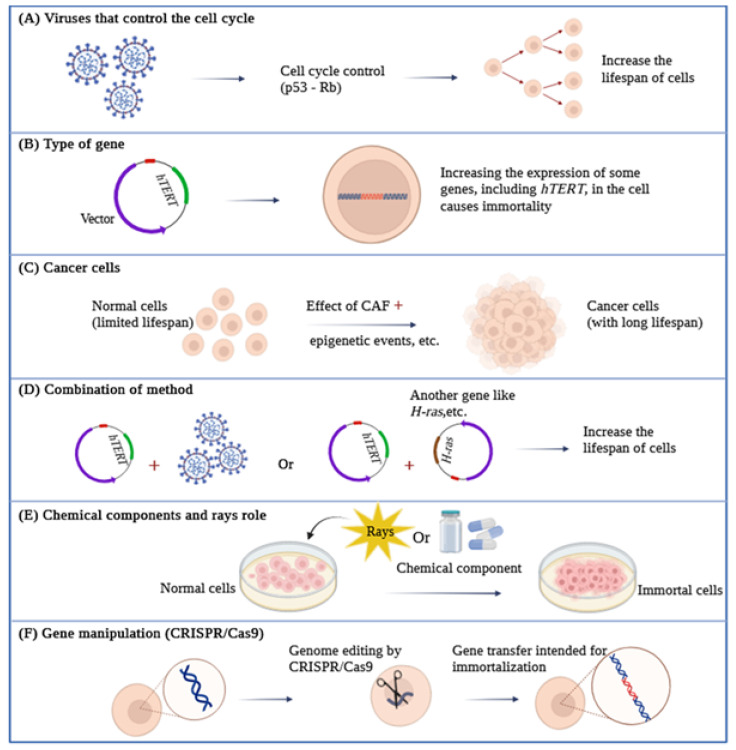
An overview of various types of cell immortalization methods.

**Table 1 life-14-00417-t001:** Types of immortalized cells along with some of their acquired characteristics.

	Immortal Cell Type	The Method of Immortality	Strategy	The Effectiveness of the Method	Gene Transfer Method	Immortal Cell Characteristics	Immortality Result	Ref.
1	Human vocal fold epithelial cells	Increased *hTERT* gene expression	Expression of the catalytic subunit of telomerase	Most effective and safe in most cell types	Retrovirus	They maintained their phenotypes with almost identical genotypes in cellular pathways and functioned properly in relation to ion and protein transport and cell signaling. They also maintained the ability of stable reproduction for more than 8 months.	- The first generation of immortalized hVFE cells obtained from the luminal surface of the true VF. - The immortalized hVFE can be easily frozen, stored, and recovered. - It can be shipped to other research institutions and widely used for various *in vitro* research in laryngeal field.	[30]
2	Primary canine corneal epithelial cells	SV40 T antigen	Induction of viral oncogenes that inactivate cell cycle proteins	Usually effective but not safe (viral gene)	Lentivirus	They maintained their biological characteristics and had a stronger proliferation capacity than normal cells. They also maintained their diploid karyotype and serum-dependent ability.	- The CCEC-SV40T line was successfully established. - Can be used for *in vitro* studies, such as research on corneal diseases or drug screening.	[166]
3	Yak rumen epithelial cells	Increased hTERT gene expression + SV40 T antigen	Expression of the catalytic subunit of telomerase + Induction of viral oncogenes that inactivate cell cycle proteins	A combination of methods is usually effective, but there is caution in using viral genes	Lentivirus	They maintained the morphological and functional characteristics of the primary cells and also had functions related to the normal transport and absorption of short-chain fatty acids. Cell proliferation and karyotype were normal.	- The immortalized cell line SV40T-YREC-hTERT was established for the first time. - SV40T-YREC-hTERT provided an experimental model for studying the biological function of the yak rumen epithelium *in vitro*.	[167]
4	Human retinal pigment epithelial RPE-1 cells	Increased *hTERT* gene expression	Expression of the catalytic subunit of telomerase	Most effective and safe in most cell types	ERT2-Cre-ERT2AAVS1 integration plasmid	Life expectancy increased. It does not have transformed phenotypes and has a stable and normal karyotype.	- The hTERT RPE-1 ERT2-Cre-ERT2 cell line was established. - Is versatile in gene editing and can facilitate future functional studies of genes and the human genome.	[31]
5	HUVECs	Increased *hTERT* gene expression	Expression of the catalytic subunit of telomerase	Most effective and safe in most cell types	Lentivirus	Cells showed longer lifespan and maintained endothelial characteristics. They expressed the factors CD31, VEGFR-II, and alpha5 integrin. Antitumor immunity was also confirmed.	- This study is the first to confirm the antitumor immunity of hTERT-immortalized HUVECs.	[168]
6	Human bone marrow mesenchymal stem cells	Increased *hTERT* gene expression	Expression of the catalytic subunit of telomerase	Most effective and safe in most cell types	Retrovirus	The restoration of telomerase activity, the increase in the life span of the cells, the characteristics of the stem cells of self-renewal, and the ability to differentiate into the mesoderm-type cell lineage were preserved.	- Telomerized hMSC lines maintain long-term self-renewal and differentiation capacity. - hMSC-TERT cell lines were capable of forming bone, bone-marrow supporting stroma, and adipocytes when transplanted subcutaneously in immune-deficient mice.	[169]
7	Human fetal hepatocytes	Increased *hTERT* gene expression + SV40 T antigen + E7	Expression of the catalytic subunit of telomerase + overexpression of certain genes for immortalization	A combination of methods is usually effective, but there is caution in using viral genes	Vector + *E. coli* plasmid	A stable cell line was obtained from human fetal liver cells that was able to secrete albumin–urea and consume glucose.	- A conditional human fetal hepatocytes cell line with mesenchymal characteristics was established.	[170]
8	Human neural stem cells	Expression of the *v-myc* gene			Retrovirus	It produces stem cells with increased proliferation capacity and they do not change shape in laboratory conditions and are not tumorigenic in the body.	- This line having been established using a much lesser mutated and better characterized c-myc mutant.	[95]
9	Ligamentocytes derived from human anterior cruciate ligament	SV40 T antigen	Induction of viral oncogenes that inactivate cell cycle proteins	Usually effective but not safe (viral gene)	Vector transfection	Transfected ligamentocytes maintain the phenotype and vital properties of the cell.	- SV40-transfected ligamentocytes express normal tendon components. - However, some differences, such as distinct cytoskeletal changes and limited survival in long-term 3D cultures, could be demonstrated. - It follows that tissue engineering (TE) with transfected SV40 cells is only possible for limited culture periods.	[171]
10	Bone marrow mesenchymal stromal cells	Increased *hTERT* gene expression	Expression of the catalytic subunit of telomerase	Most effective and safe in most cell types	Lentivirus	Proliferated longer than wild-type BMSCs. Phenotype and karyotype were not significantly different from non-transfected cells. The cells also maintained the normal morphology and neural differentiation characteristics of stem cells when cultured in induction media.	- An hTERT-BMSCs/Tet-on/GAL cell line was constructed using a single Tet-on-inducible lentivirus system. - Subsequent experiments demonstrated that the secretion of rat GAL from hTERT-BMSCs/Tet-on/GAL was switched on and off under the control of an inducer in a dose-dependent manner.	[156]

## Data Availability

Not applicable.

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
