# Peer review of "Cell Immortality: In Vitro Effective Techniques to Achieve and Investigate Its Applications and Challenges"

_life, 2024, doi:10.3390/life14030417_

Round 1

Reviewer 1 Report (Previous Reviewer 1)

Comments and Suggestions for Authors

Manuscript still requires some proof-reading and corrections to grammar, sentence structure and punctuation. A few examples below.

This marked the first instance of such a phenomenon. [17, 18]. Prior………….There is a punctuation before citation.

behavior of lacks tumor cells was different………Should be “behavior of Lacks’s tumor cells was different.”

This was the first successful attempt to immortalize and proliferate human cells in laboratory conditions, and the rapid growth and reproduction of HeLa cells made them widely used…..Rephrase for better clarity

Shortening of telomeres is a mechanism for suppressing tumor by activating cellular aging signals.---------should read Shortening of telomeres is a presumed tumor suppressor pathway  mediated, in part, by activation of cellular aging signals.

Human primary cells that have been immortalized by increasing hTERT expression such as fibroblasts [122], choroidal melanocytes [31], endothelial cells [123], dermal keratinocytes, mammary epithelial cells, osteoblasts, and pancreatic cells [124]……Should read “Human primary cells that have been immortalized by increasing hTERT expression in fibroblasts [122], choroidal melanocytes [31], endothelial cells [123], dermal keratinocytes, mammary epithelial cells, osteoblasts, and pancreatic cells [124].

About telomere shortening and proper checkpoint activities, the shortening of telomeres can be regarded as an anti-cancer mechanism only in case of a proper activity of the cellular checkpoints such as P53 and RB.-----Rephrase for clarity.

In the field of overexpression exogenes for immortalization, some of them only have the ability to immortalize a series of specific primary cells (such as HOX family genes and others, such as c-Myc, have oncogenic activity, so using them may effect safety the process.------ Rephrase for clarity.

Comments on the Quality of English Language

See above comments.

Author Response

Thank you very much for your comments. We have made the changes you want (Turquoise).

Reviewer 2 Report (Previous Reviewer 2)

Comments and Suggestions for Authors

acceptable for publication now

Author Response

Thank you very much for your time.

This manuscript is a resubmission of an earlier submission. The following is a list of the peer review reports and author responses from that submission.

Round 1

Reviewer 1 Report

Comments and Suggestions for Authors

There are numerous serious concerns with this manuscript. To list a few;

1) Extensive revision of English, content and discuss is required.

2) The authors on multiple occasions misrepresent citations either not appropriately discussing them or by providing citations for that are completely unrelated to their statements. For example, "Additionally, to help reduce the num-ber of aging cells in individuals with premature aging syndrome ...... [30]" sounds as if studies were conducted in individuals when the cited work is completely in vitro and shows that immortilization simply stabilized an in vitro phenotype. In another example, "Notably, these cells have been employed safely in clinical trials focused on treating brain injuries....[37] ,"  citation 37 says nothing about clinical trials or brain injury. This happens multiple times throughout the manuscript.

3) Throughout the manuscript punctuation needs to be revised. There are numerous instances where a period is placed before a citation. For example, "This arrangement inhibits the identification of linear chromosome ends as double-stranded DNA breaks. [45]. ."

4) All the figures provided throughout the manuscript are blurry to some extent. 

5) Incorrect capitalization of words occurs throughout. For example, "Conversely, Garcìa-Mesa and collaborators successfully immortalized microglial cells for the purpose of studying the latency and regulation of the Human Immunodefi-ciency Virus (HIV) in the Central Nervous System (CNS). ."

6)Sections throughout the manuscript are presented as a simple list of things and not as a discussion, most prominent in throughout section 4. 

7) Some sentences and sections are nonsense or difficult to comprehend. For example, "According to the explanations provided, we understand that in these methods for immortalization, finally, the sequence (mostly of viral origin) is transferred to the cell ge-nome that was not naturally present in the genome, so such manipulation of the cell ge-nome cannot produce safety results because as It was said that these pathways target most of the key regulators of the cell, which can be concluded that the use of onco genes, alt-hough considering the intended use, cannot be a completely safe method. ." and "The unique properties of hTERT-immortalized cells, as discussed, make them a suitable replacement for other methods of immortaliza-tion to develop immortal cell lines and a valuable tool for studying cellular function in laboratory and in vivo conditions. ."

Overall this manuscript requires a complete overhaul, editing and in depth discussion with care citations that accurately represent prior work. 

Comments on the Quality of English Language

See comments and suggestions above. 

Reviewer 2 Report

Comments and Suggestions for Authors

In this review, the authors describe in a comprehensive way techniques to achieve cells immortalization for in vitro laboratory use. The review is important and thorough, I have found minor issues that require their attention.

1.    English languish should be revised by an English speaker professional editor, as an example there are several phrases which include: "In a study….." – This should be mended appropriately.

2.    P. 2 second raw: should not be placed here and the term "in vitro" should be added.

3.    Table 2 can be fused to table one, just by adding another column to table 1- an "example" column.

4.    On p. 7: add an explanation regarding the connection between telomere shortening and proper checkpoint activities. Please write that the shortening of telomeres can be regarded as an anti-cancer mechanism only in case of a proper activity of the cellular checkpoints such as P53 and RB.

5.    Figure 4. Remove the "methods of immortalization" column.

6.    Instead of Table 2 the whole review may be summarized in a new Table summarizes all of the examples mentioned in the text in the same order.

7.    When describing the biology of hTERT it should be clearly stated that it is not an oncogene.

8.    P. 12, please erase the first section (including refs 126, 127) as it is a repetition.

9.    On page 14 when describing the drawbacks of using cell lines that have been immortalized it should be mentioned that these cell may acquire many mutations during prolonged growth which mask their authenticity.

Comments on the Quality of English Language

Many phrases should be re write, please send the manuscript to an English speaker editor.

Reviewer 3 Report

Comments and Suggestions for Authors

Chalak et al. put together a comprehensive review technique involved in cell immortalization. However, I have a few comments to improve the overall comprehension of this work.

  • In the introduction, I need help understanding the term M1 phase. What does that mean? Does cell cycle arrest not occur in G1 (before S) or G2 (before mitosis)? Does M1 refer to early mitosis? Please specify
  • The meaning of iPSCs cells is not mentioned at the end of page 3 (Induced pluripotent stem cells (iPSCs)
  • Page 4. You start the paragraph with "additionally." Which technique do you mention here? There is a considerable subject change here; I suggest providing background knowledge before that.
  • All your figures appear in shallow resolution. Please provide high-quality images.
  •  Table 1 needs a legend. What are CAFs? What are miRs? miRNAs? Rays? Is X-Rays? I would not name column 3 as impact rate; maybe the term "results" or other terminology would be more appropriate.